# Contribution of Hydrolysis and Drying Conditions to Whey Protein Hydrolysate Characteristics and In Vitro Antioxidative Properties

**DOI:** 10.3390/antiox11020399

**Published:** 2022-02-16

**Authors:** Thanyaporn Kleekayai, Aileen O’Neill, Stephanie Clarke, Niamh Holmes, Brendan O’Sullivan, Richard J. FitzGerald

**Affiliations:** Department of Biological Sciences, University of Limerick, V94 T9PX Limerick, Ireland; thanyaporn.kleekayai@ul.ie (T.K.); aileen.oneill@ul.ie (A.O.); stephanie.clarke@ul.ie (S.C.); niamh.holmes@ul.ie (N.H.); brendan.osullivan@ul.ie (B.O.)

**Keywords:** whey, hydrolysate, drying, freeze-dry, spray-dry, hydrolysis condition, pH-stat, free-fall, antioxidant

## Abstract

During the generation of functional food ingredients by enzymatic hydrolysis, parameters such as choice of enzyme, reaction pH and the drying process employed may contribute to the physicochemical and bio-functional properties of the resultant protein hydrolysate ingredients. This study characterised the properties of spray- (SD) and freeze-dried (FD) whey protein hydrolysates (WPHs) generated using Alcalase^®^ and Prolyve^®^ under pH-stat and free-fall pH conditions. The enzyme preparation used affected the physicochemical and antioxidative properties but had no impact on powder composition, morphology or colour. SD resulted in spherical particles with higher moisture content (~6%) compared to the FD powders (~1%), which had a glass shard-like structure. The SD-WPHs exhibited higher antioxidative properties compared to the FD-WPHs, which may be linked to a higher proportion of peptides <1 kDa in the SD-WPHs. Furthermore, the SD- and FD-WPHs had similar peptide profiles, and no evidence of Maillard reaction product formation during the SD processing was evident. The most potent in vitro antioxidative WPH was generated using Alcalase^®^ under free-fall pH conditions, followed by SD, which had oxygen radical absorbance capacity and Trolox equivalent (TE) antioxidant capacity values of 1132 and 686 µmol TE/g, respectively. These results demonstrate that both the hydrolysis and the drying process impact the biofunctional (antioxidant) activity of WPHs.

## 1. Introduction

Whey-based ingredients are widely used due to their high protein quality (being a rich source of essential and branched-chain amino acids) and digestibility [1,2,3,4]. Whey proteins and their derivatives (including hydrolysates and peptides) have demonstrated numerous health benefits, e.g., associated with prevention and management of conditions such as cardiovascular disease, type II diabetes mellitus, cancer, inflammation, sarcopenia, obesity, etc. [1,2,3,4,5,6,7,8].

Oxidative stress is reported to play a major role in many health conditions. Likewise, many of the health-promoting effects of whey proteins are attributed to their antioxidative properties. Whey proteins are reported to act as potent antioxidants by modulating a range of redox biomarkers, e.g., glutathione, superoxide dismutase, glutathione peroxidase, catalase, nitric oxide, malondialdehyde and reactive oxygen species [6,7]. The potent antioxidative properties of whey proteins has been associated with their hydrophobic amino acid residues [9], their ability to act as a source of cysteine residues to synthesise glutathione, a potent intracellular antioxidant compound and their ability to form Maillard reaction products in food systems [3]. Pre-digested whey proteins and their peptides are more effectively absorbed in the gut and are more rapidly transported into the bloodstream compared to undigested whey proteins [7,8]. Furthermore, whey protein-derived hydrolysates and peptides are reported to possess higher physiological and biological potential compared to intact whey proteins [1,6,9,10].

Enzymatic hydrolysis has been widely used for modification of the techno- and bio-functional properties of food proteins. These characteristics are often impacted by the proteolytic enzyme used and the hydrolysis conditions employed, including enzyme:substrate (E:S), pH of the reaction, operating temperature, type of salt and ionic strength [11,12]. The pH of the reaction is considered as one of the most important parameters during enzymatic hydrolysis. Several studies have demonstrated that hydrolysis, with and without pH regulation, influences the physicochemical properties, peptide profiles and bio-functional properties of whey protein hydrolysates [13,14,15,16,17].

During manufacture, functional food ingredients undergo a range of thermal processing steps to transform liquid streams into more stable powder forms. Drying is a key process to achieve stable powder ingredients, which, in itself, may influence the physicochemical and biological properties of the final ingredients. Freeze-drying (FD) and spray-drying (SD) are the most common processes used to obtain dry powders at both the lab- and industrial-scale. FD is a dehydration process based on the sublimation of a frozen product at low temperature. It is, thus, frequently used for heat-sensitive materials and is recognised as one of the best methods to yield high-quality dry ingredients due to its limited ability to induce different chemical and microbiological reactions [18]. Furthermore, FD has been shown to fully preserve the nutritional properties, e.g., the nutritional properties of freeze-dried camel milk were similar to that of fresh milk [19]. However, this drying technique has a lower throughput, a higher capital cost and a longer operation time compared to SD. On the other hand, spray-drying provides a larger throughput with a shorter operation time in a single step. However, exposure to high temperatures (>100 °C) during SD may lead to chemical modifications, particularly Maillard reactions. This may modify the nutritional value of spray-dried products, e.g., leading to reduced amino acid bioavailability, which in turn may influence the bio-functional properties of spray-dried proteinaceous ingredients.

Functional food ingredients derived from dairy protein hydrolysates generally contain a mixture of short-chain polypeptides and free amino acids, which can be readily susceptible to the Maillard reaction during thermal processing. The Maillard reaction is based on the reaction of free amino groups (-NH_2_ groups of free amino acid residues, peptides and proteins) and carbonyl groups of reducing sugars [3]. Studies have demonstrated that Maillard reaction products derived from whey protein ingredients and their hydrolysates through conjugation with carbohydrates resulted in an enhancement in their bio-functional properties, mainly associated with their antioxidative properties. For instance, an increase in 2,2′-azinobis (3-ethylbenzothiazoline-6-sulfonic acid) cation radical (ABTS^•+^) and hydroxyl radical scavenging activity, ferric reducing powder (FRAP), oxygen radical absorbance capacity (ORAC), inhibition of cellular reactive oxygen species and upregulation of glutathione level in hepatocyte (HepG2) cell lines has been reported in whey protein-derived sugar/carbohydrate conjugates [20,21,22,23]. Moreover, an enhancement in the techno-functional properties of the conjugates has also been reported, e.g., in the heat stability, water holding capacity, foaming capacity and stability, and in the emulsion activity and stability [20,21,24]. 

However, there is limited information regarding the impact of SD vs. FD on the characteristics of whey protein hydrolysates generated under different hydrolysis conditions. Therefore, the aim of the present study was to investigate the effects of whey protein hydrolysis using Alcalase^®^ and Prolyve^®^ generated under pH-stat and free-fall pH conditions on hydrolysate physicochemical and in vitro antioxidative properties. In addition, the impact of SD vs. FD processes on the hydrolysate properties was also examined.

## 2. Materials and Methods

### 2.1. Materials

Whey protein concentrate containing ~80% (*w*/*w*) protein (WPC80) was kindly provided by Carbery (Balineen, Cork, Ireland). The food-grade enzyme preparations used were Alcalase^®^ 2.4 L (2.4 Anson units/g) and Prolyve1000^®^ (≥3000 U/g protein), obtained from Sigma-Aldrich (Dublin, Ireland) and Soufflet Biotechnologies (Caen, France), respectively. Trifluoroacetic acid (TFA), Trolox and 2,2′-azobis (2-amidinopropane) dihydrochloride (AAPH) were purchased from Sigma-Aldrich. High-pressure liquid chromatography- (HPLC) and mass spectrometry (MS)-grade water and acetonitrile (ACN) were provided by Fisher Scientific (Dublin, Ireland). 2,4,6-Trinitrobenzenesulfonic acid (TNBS) was obtained from Pierce Biotechnology (Medical Supply, Dublin, Ireland). Food-grade sodium hydroxide (NaOH) was obtained from Micro-Bio Ireland Ltd. (Cork, Ireland).

### 2.2. Generation of Whey Protein Hydrolysates (WPHs)

The WPHs were prepared using the method previously described in Kleekayai et al. [15] with some modifications. In brief, a 10% (*w*/*v*, protein) WPC80 suspension was reconstituted (~3.5 L) in spring water (Ishka, Limerick, Ireland) and pre-equilibrated at 50 °C for 1 h. The pH of the solution was then adjusted to pH 7.0 and 9.0 for pH-stat (ST) and free-fall (FF) pH conditions, respectively, using 2 M NaOH. The enzyme, Alcalase^®^ 2.4 L (Alc) or Prolyve1000^®^ (Plv), was then added at 1% (*v*/*w*) on a protein basis to initiate the hydrolysis reaction. The reaction was carried out at 50 °C for 4 h. For the ST conditions, the hydrolysis reaction was maintained at pH 7.0 throughout the hydrolysis process. While for the FF conditions, the pH was not adjusted during the reaction; however, it was adjusted to pH 7.0 following 4 h incubation. The hydrolysis reaction in both cases was terminated by heating in a water bath at 80 °C for 25 min.

### 2.3. Drying of the WPHs

Spray- and freeze-drying was employed to generate powder samples of the WPHs. Hydrolysate samples were spray-dried using a Büchi Mini Spray Dryer B-290 (Flawil, Switzerland) with inlet and outlet temperatures of 180 °C and 80–90 °C, respectively, at a feed rate of 300 L/h. During freeze-drying, the samples were firstly frozen at −20 °C overnight prior to freeze-drying using a FreeZone 18 L Labconco freeze-drier (Kansas City, MO, USA). The frozen samples were freeze-dried at −51 °C at a pressure <0.133 mBar for ~96 h. The dried samples were packed in air-tight containers and stored at −20 °C until used.

### 2.4. Compositional Analysis

The moisture and crude protein/protein equivalent contents of the WPHs obtained during SD, and FD were determined using the hot-air oven and Kjeldahl methods of the International of Dairy Federation (IDF) 026:2004 and IDF 020-1:2014, respectively. For crude protein determination, a conversion factor of 6.38 was used for milk products (IDF 020-1:2014). Ash content was analysed using the standard method of the AOAC (Method 930.30 [25]). Water activity (a_w_) was analysed using an AQUA lab water activity analyser (METER Group, Inc., Pullman, WA, USA) at 20 ± 1 °C; the instrument was pre-equilibrated with 0.5 M KCl prior to analysis.

### 2.5. Measurement of Colour and Maillard Reaction Product Formation

The colour of the SD and FD powders was measured using a colourimeter (Spectrophotometer CR-600d, Konica Minolta Inc., Osaka, Japan). Measurements were performed in triplicate, and the results were expressed using the CIELAB colour space as *L**, *a** and *b** representing lightness, red-green and yellow-blue axes, respectively. The browning index of the powder samples was calculated based on the CIElab colour space values according to Schong and Famelart [26].

For Maillard reaction product determination, the samples were prepared as 0.5% (*w*/*v*) protein suspensions in distilled water prior to measurement of non-fluorescent Maillard reaction product intermediates at absorbance values of 280 and 294 nm. The browning associated with late Maillard reaction and advanced Maillard products was measured at an absorbance of 420 nm and at fluorescence emission/excitation wavelengths of 330/420 nm, respectively [27,28].

### 2.6. Analysis of Surface Morphology

The surface morphology of the SD and FD samples was analysed using a Hitachi SU70 scanning electron microscope (SEM; Hitachi High Technologies Corp., Tokyo, Japan) with an electron beam acceleration of 10 kV. The samples were coated with gold/palladium at 20 mA under vacuum prior to analysis. The images were captured at magnifications of 300 and 1000×.

### 2.7. Characterisation of WPH Physicochemical Properties

The extent of hydrolysis was analysed using the TNBS assay, and the results were expressed as the degree of hydrolysis (DH), which characterises the extent of free amino groups released during enzymatic hydrolysis compared to the amino group content in a corresponding unhydrolysed sample, as described previously [15]. Molecular mass distribution and reverse-phase (RP) profiles were analysed using a Waters 600 high-performance liquid chromatography (HPLC) and an Acquity ultra-performance liquid chromatography (UPLC) system (Waters, Dublin, Ireland), respectively. The HPLC was coupled with a gel permeation (GP) column, TSK G2000 SW (600 mm × 7.5 mm I.D.; TOSOH Bioscience, San Francisco, CA, USA) connected to a TSKGEL SW guard column (75 mm × 7.5 mm I.D.; TOSOH Bioscience). While the UPLC was equipped with an Acquity UPLC BEH C18, 130 Å column (2.1 mm × 50 mm × 1.7 mm; Waters, Milford, MA, USA) connected with an Acquity BEH C18 Vanguard pre-column (Waters, Milford, MA, USA). The GP-HPLC and RP-UPLC were performed as described by Kleekayai et al. [15].

### 2.8. Assessment of In Vitro Antioxidative Activity

The oxygen radical absorbance capacity (ORAC) and the Trolox equivalent antioxidant capacity (TEAC) of the SD- and FD-WPHs were assessed using the methods previously described by Kleekayai et al. [15]. The ferric reducing antioxidant power (FRAP) assay was carried out as described by Cermeño et al. [29]. Each sample was analysed in triplicate (*n* = 3). The results were expressed as µmol Trolox equivalents per gram of SD/FD powder (µmol TE/g).

### 2.9. Statistical Analysis

All analyses were conducted in triplicate (*n* = 3), and the results are presented as the mean ± standard deviation (SD). All statistical analyses were performed using SPSS Statistics V. 26 (IBM Inc., Chicago, IL, USA). Mean value comparison between treatments and/or groups were analysed using one-way or two-way analysis of variance (ANOVA) with a Tukey’s test. A dependent (paired) sample T-test was used to compare mean values of the SD vs. FD samples. Significance was determined at *p* < 0.05.

## 3. Results and Discussion

### 3.1. Characteristics of the WPH Powders

The properties of the SD-WPHs are presented in Table 1. The moisture content of the different SD samples, including WPC80, were not significantly different (*p* > 0.05) and ranged between 5.3–6.2%. The moisture contents of the SD-WPHs (5.39–6.25%; Table 1) were higher (*p* < 0.05) than that of the FD-WPHs, which ranged between 0.96 ± 0.15 and 1.55 ± 0.52% (data not shown), this is in agreement with previous studies [30,31]. The different moisture contents in the SD vs. FD powders may be associated with the different principles in the dehydration processes and the conditions employed. The higher moisture content in the SD powders may be linked to the smaller particle size and larger surface area, than in the FD powers, which may lead to increased hydroscopicity [31]. The lower residual moisture content in the FD powder may provide it with a greater long-term stability compared to the SD powder [32]. Similar protein equivalent contents in the SD- and FD-WPHs were observed which were in the range of 77.09–79.34% for the SD-WPHs (Table 1) and between 78.18 ± 0.33% to 79.86 ± 0.39% for the FD-WPHs (data not shown). The results indicated that the drying process used (SD vs. FD) did not influence the protein equivalent content of the WPHs on a dry weight basis. Chen et al. [33] and Wang et al. [34] did not observe any impact of SD vs. FD on the moisture and protein contents of egg white and soybean protein hydrolysates. However, it should be noted that parameters associated with the SD conditions, i.e., feed pump rate, airflow rate, inlet and outlet temperatures, as well as the feed concentration/total solids and feed temperature, can impact the characteristics of SD powders. Higher inlet and outlet temperatures and lower feed flow rates can, e.g., contribute to a decrease in moisture content and water activity, while a higher total solids can result in a lower moisture content [18,32].

A higher ash content was present in all WPHs compared to intact WPC80; this was related to the pH adjustments during ST7 and after FF9 hydrolysis. This led to a lower protein equivalent content in the WPHs (77.1–79.3%) compared to the protein content in WPC80 (81.8%). The a_w_ of the SD-WPH samples ranged between 0.33 and 0.43, which was similar to the a_w_ of WPC80 (0.39); these values are within the recommended values (a_w_ ≤ 0.6) to ensure microbial stability [35]. The DH indicates the extent to which the whey proteins were digested using Alcalase^®^ and Prolyve^®^. The DH is an important parameter as it can influence the techno- and biofunctional properties of protein hydrolysates. The DH can be impacted by a range of parameters, e.g., substrate type, enzyme specificity, total solids, E:S, reaction pH, incubation time and temperature [11,12]. In this study, the DHs of the WPHs were between 9.5–13.2% following 4 h incubation (Table 1). In the case of Alcalase^®^ hydrolysis of WPC, it was noted that a significantly higher DH was achieved during the ST conditions in comparison to the samples generated under FF conditions. No significant difference in DH was observed in the Prolyve-WPHs during ST and FF hydrolysis. Similar DH values in the WPHs generated under FF and ST conditions was previously reported during Alcalase^®^ hydrolysis of whey protein isolate (WPI) [17] and during WPC80 hydrolysis using papain [16], FlavourPro^®^ Whey and Debitrase^®^ HYW20 [15]. Eberhardt et al. [36] reported that pH significantly influenced the DH value of WPC80 hydrolysed with Alcalase^®^. The highest DH value (23.1%) was achieved when hydrolysis was carried out using ST conditions at pH 9.0, an E:S of 2% (*w*/*w*) and 10% (*w*/*v*) WPC80, while ST at pH 7.0 yielded the lowest DH value (16.9%). The decrease in pH during FF hydrolysis modified the kinetics and cleavage specificity of Alcalase^®^ [13], which in turn led to a lower extent of hydrolysis [15]. In the present study, the pHs of the FF9 hydrolysis reaction decreased to ~pH 6.6–6.7 following 4 h incubation at 50 °C (data not shown). Both enzyme preparations used for WPC hydrolysis herein are serine alkaline proteinase derived from *Bacillus licheniformis*. However, glutamyl endopeptidase (GE) activity which is present as a side-activity in Alcalase^®^ was not found in Prolyve^®^ [9,37]. This may contribute to the different properties for the Alcalase^®^ and Prolyve^®^ hydrolysates.

### 3.2. Colour and Maillard Reaction Products

The SD and FD samples were characterised with respect to their colour in order to investigate the effect of the drying process on their physical characteristics. The SD samples had lower lightness values compared to the FD-WPHs and WPC80 (Figure 1a). A similar result was also observed in the case of SD and FD sodium caseinate [31]. The yellowness (*b**) and browning index values of the SD samples were also lower than that of the FD-WPHs and WPC80 (Figure 1c,d, respectively). Meanwhile, the redness (*a**) value of the SD and FD samples was enzyme-dependent (Figure 1b). 

A reduction in *L** value was previously reported following the conjugation of WPC, WPH and WPI with polysaccharides due to the formation of Maillard reaction products [22,27]. However, there was no evidence of the presence of intermediate and late Maillard reaction products that may occur during SD as a result of high heat treatment (Appendix A) or of a reduction in free amino groups in the SD powders as a result of the Maillard reaction in comparison to the FD powders (Appendix A). A lower yellowness value in the SD compared to FD powder was also observed in egg white protein hydrolysates [33]. It is evident that the appearance of WPHs depends on the drying process, i.e., FD vs. SD. 

### 3.3. Surface Morphology

SEM was carried out to examine the surface morphology of the SD and FD powders. Distinct morphological differences and particle sizes can be observed between the SD and FD samples, as shown in Figure 2. SD produced hollow spherical particles with some irregular shrunk and cracked particles, as observed in the SD-WPHs (Figure 2) and in the WPC80 used for the manufacture of the WPHs (Appendix A). This phenomenon has been attributed to the rapid evaporation of moisture during the spray-drying process and the formation of a protective film on the droplet surface [34]. 

The particle sizes of the SD powder particles were heterogeneous with a distribution of discrete micron to submicron particles. Atomisation during SD produced small droplets resulting in spherical structures in the SD powder. In contrast, the FD powders had a glass shard-like structure due to the sublimation process (Figure 2). These results are in agreement with previous studies showing the effect of SD and FD on the microstructure of whey protein [30] and egg white protein hydrolysates [33]. It was reported that sample feed rate, total solids and air inlet temperature impact the particle size in SD powders. Meanwhile, the characteristics of FD powders depend on the freezing conditions [18].

Furthermore, powder particle size can have an impact on powder solubility, with smaller particles being associated with greater solubility. This is due to a higher surface area being available for hydration in small particles as opposed to larger particles [34]. Powder solubility during reconstitution can also affect food formulations as well as the techno-and biofunctional properties of protein ingredients [33,38,39]. Nonetheless, previous studies did not report on the impact of the drying process (SD vs. FD) on the solubility of whey and egg white protein hydrolysates [30,33]. It seems that the enzyme preparation and hydrolysis conditions used did not alter the morphological characteristics of the SD and FD powders.

### 3.4. Molecular Mass Distribution and Peptide Profiles

The molecular mass distribution profiles of the SD- and FD-WPHs and WPC80 are displayed in Figure 3. All the WPHs displayed higher proportions of low molecular mass components between 1–5 and <1 kDa compared to WPC80. Although the extent of hydrolysis of the WPHs were within a narrow range (9.5–13.2% DH), significant differences in the molecular mass distribution profiles were observed. The Alcalase hydrolysates (Alc-ST7 and Alc-FF9) had a higher proportion of lower molecular mass components <1 kDa (~60%) compared to the Prolyve hydrolysates (Plv-ST7 and Plv-FF9 having 42–44% < 1 kDa). This may be linked to the cleavage specificity of Alcalase^®^, resulting in an hydrolysate containing small molecular mass peptides [40]. However, the molecular mass distribution profile of Alc-FF9 displayed a higher proportion of molecular mass components > 10 kDa compared to Alc-ST7, which may be associated with the lower DH value of Alc-FF9 (Table 1). In the case of Prolyve^®^, the hydrolysis conditions (ST7 vs. FF9) did not appear to have a major impact on the molecular mass distribution profile. This concurs with the observation that these samples had similar DH values (Table 1). It is well documented that the bioactive properties, e.g., antioxidative properties, of protein hydrolysates are associated with low molecular mass components [6,9,41]. Therefore, the differences in molecular mass distribution profiles generated between the WPHs herein may contribute to their antioxidative potencies.

There were some differences in the molecular mass distribution profiles of the WPHs depending on the drying process used (SD vs. FD). In general, the SD-WPHs contain a greater amount of low molecular mass components (<1 kDa) than the FD-WPHs, particularly in the case of Alcalase hydrolysates. For instance, the Alc-ST7 contained 58% and 53% of peptides <1 kDa in the SD and FD samples, respectively (Figure 3). These results are comparable to a previous study by Shi et al. [42], who reported that an SD silkworm pupae protein hydrolysate had a higher proportion of peptides < 1 kDa (83.6%) compared to the corresponding FD powder (77.1%). In contrast, no major differences were reported in the molecular mass distribution profiles of egg white protein hydrolysates obtained following SD and FD [33]. The effect of SD on the molecular mass distribution profiles of the WPHs may be associated with the thermal sensitivity of residual intact whey proteins. Exposure of thermally-labile proteins to the air/liquid interface at the droplet surface during atomisation may lead to aggregation resulting in poorer solubility in SD powders [32]. In addition, the thermal treatments associated with the SD process may lead to activation of any residual enzyme activity in the hydrolysate resulting in a lower increase in DH.

Similar to the molecular mass distribution profiles, the RP-UPLC chromatograms also showed the degradation of intact whey proteins, including β-lactoglobulin (β-Lg), α-lactalbumin (α-La), bovine serum albumin (BSA) and glycomacropeptide (GMP) during hydrolysis with Alcalase^®^ and Prolyve^®^, as shown in Figure 4. Regarding the impact of the drying process on the peptide profile, there were no major discernible differences between the peptide profiles of the SD-and FD-WPH samples. However, as expected, different peptide profiles were observed depending on the enzyme preparation used. Despite both enzyme preparations being derived from *B. licheniformis*, Prolyve^®^ does not contain GE activity, which can be found in Alcalase^®^, as described elsewhere [37]. 

The results herein agree with a previous study reporting distinct RP-HPLC profiles during WPC80 hydrolysis using Alcalase^®^ and Prolyve^®^, when generated under ST pH 7.0 conditions [37]. However, it was noted that the hydrolysis conditions (ST7 vs. FF9) did not show a major impact on the peptide profiles of the WPHs obtained with both enzyme preparations, while the DH value of Alc-FF9 was significantly lower (*p* < 0.05) than that of the Alc-ST7 (Table 1). On the contrary, Carvalho et al. [17] reported different peptide profiles for WPI hydrolysis using Alcalase^®^ following ST and FF starting at pH 8.5. Butré et al. [13] demonstrated the influence of pH (between pH 7.0–9.0) during WPI hydrolysis on the selectivity of Alcalase^®^, which in turn affected the peptide profiles of the WPHs generated. A number of parameters may contribute to different findings, e.g., differences in the whey protein substrate used (WPC80 vs. WPI), E:S and the specific enzyme activity used during the generation of hydrolysates. Commercial enzymes are generally crude enzyme preparations; therefore, enzyme activity may vary depending on the supplier and the batch of the enzyme [9,12]. A number of studies have characterised protein hydrolysis using Alcalase^®^; however, to the best of our knowledge, this is the first study demonstrating the effect of hydrolysis conditions and the drying process on Prolyve-derived WPC hydrolysate properties.

### 3.5. In Vitro Antioxidative Activity

Measurement of in vitro antioxidative activity was used as an indicator of the impact of the hydrolysis conditions and the drying process on the bioactive properties of the hydrolysates. The in vitro antioxidative activity of the WPHs was assessed using three different biochemical-based assays, i.e., the ORAC, TEAC and FRAP assays, and the results obtained are shown in Figure 5. As expected, the ORAC (410.87–1132.01 µmol TE/g) and TEAC (381.63–686.46 µmol TE/g) values for the WPHs were significantly higher than WPC80 (<100 µmol TE/g, Figure 5a,b). Moreover, it was noted that the ORAC and TEAC values of the SD- and FD-WPHs followed a similar pattern and were enzyme-dependent. Hydrolysis using Alcalase^®^ resulted in higher ORAC and TEAC values compared to WPC hydrolysis with Prolyve^®^. These results concur with those reporting higher antioxidant activities in Alcalase^®^-generated WPHs as opposed to other enzyme preparations [36,43,44]. On the other hand, hydrolysis of WPC80 using both Alcalase^®^ and Prolyve^®^ in this study did not show an improvement in FRAP value (9.21–15.97 µmol TE/g) compared to WPC80 (15.03 ± 1.78 µmol TE/g) (Figure 5c). This implies that the WPHs generated did not have the ability to reduce the ferric (Fe^3+^) to ferrous (Fe^2+^) ion under the test conditions. However, WPI hydrolysis using Alcalase^®^, bromelain and Neutrase^®^ were reported to exhibit significantly higher FRAP values (up to 26.9 µmol TE/g) compared to intact WPI (3.8 µmol TE/g) [44]. The different outcomes may be due to the different types of whey protein substrate and hydrolysis conditions used, particularly in the case of the Alcalase^®^ hydrolysed samples. 

The antioxidative properties of protein hydrolysates and peptides have been linked with low molecular mass peptides and their amino acid composition, especially the presence of hydrophobic amino acids (e.g., Leu and Val), sulphur-containing (Cys and Met), aromatic (Phe, Trp and Tyr) and His residues [6,41]. Alcalase^®^ and Prolyve^®^, which both contain subtilisin activity, preferentially cleave at hydrophobic amino acids [9,37,40,45]. However, hydrolysis with Alcalase^®^ resulted in a higher proportion of low molecular mass components (<1 kDa) compared to hydrolysis with Prolyve^®^ (Figure 3), potentially due to the presence of GE activity in Alcalase^®^. This may contribute to the higher antioxidant activity (ORAC and TEAC values) in the Alcalase^®^ hydrolysed samples. The results in the present study are in agreement with those previously reported, which indicated a correlation between low molecular mass peptides <1 kDa and the antioxidative properties of protein hydrolysates from different sources, e.g., WPC80, sodium caseinate and silkworm pupae protein [42,46,47,48].

Regarding the drying process and hydrolysis conditions, the SD-WPHs had significantly higher antioxidant activity (ORAC and TEAC values) compared to the FD-WPHs, except in the case of the Plv-FF9 WPHs, where the ORAC and TEAC values were similar (Figure 5). It is well documented that potent antioxidant activity is associated with chemical reactions, such as the Maillard reaction. However, no indication of Maillard reaction products and their intermediates being generated during SD were observed in the present study (Appendix A). Therefore, a possible explanation may be linked to the higher level of lower molecular mass components in the SD-compared to the FD-WPHs, as mentioned earlier. This was more pronounced in the Alcalase^®^ compared to the Prolyve^®^ hydrolysates. In addition, the results showed that Alc-FF9 obtained following SD exhibited the most potent antioxidant activity (ORAC and TEAC) among all samples generated (Figure 5), despite it having the lowest DH value (Table 1). The ORAC value of the SD Alc-FF9 was two times higher than that of its corresponding FD sample (Figure 5a). The SD Alc-FF9 sample yielded a 38.7% higher ORAC value compared to the SD Alc-ST7 WPH, whereas no significant difference was observed in both Alcalase hydrolysates obtained via FD (Figure 5a). There was no major impact of the hydrolysis conditions and drying process used on the ORAC and TEAC activities in the case of the Prolyve hydrolysates (Figure 5a,b). This may be associated with the specificity of the enzyme preparations as a result of the hydrolysis conditions employed. These findings are in agreement with a previous study showing significant variation in enzyme selectivity of *B. licheniformis* protease toward WPI as a function of pH during hydrolysis [13]. Therefore, this may explain the contribution of the hydrolysis conditions to the different antioxidative potencies of the WPHs prepared using both enzyme preparations in the present study. Spellman et al. [37] reported different peptide profiles between Alcalase^®^ and Prolyve^®^ hydrolysed WPC80 with particularly higher levels of β-Lg f(43–57) in Prolyve hydrolysates. Furthermore, they also demonstrated a direct correlation between the GE activity present in Alcalase^®^ and the level of bitterness in the sample. Therefore, it should be highlighted that the choice of enzyme not only influences the techno- and biofunctional properties of the resultant hydrolysate, but can also affect its sensory properties.

A number of studies have demonstrated the impact of hydrolysis conditions on the bioactive properties of protein hydrolysates, including WPC80 hydrolysis. For instance, our previous study demonstrated that the contribution of the hydrolysis conditions on the physicochemical and in vitro antioxidative properties was enzyme-dependent [15]. It was observed that Debitrase^®^ HYW20 hydrolysed WPC80 generated under ST conditions exhibited significantly higher ORAC and TEAC activities compared to the FF hydrolysate. However, this was not the case for FlavorPro^®^ Whey 750P generated hydrolysates. This was despite the fact that no significant impact of the hydrolysis conditions was observed in the cellular antioxidative activity against reactive oxygen species (ROS) generation in AAPH-stressed HepG2 cell lines. A similar trend was observed by Le Maux et al. [16], who showed that papain hydrolysed WPC80 generated under ST conditions at pH 7.0 had a significantly higher ORAC value compared to the WPH generated under FF conditions, even though comparable DH values and peptide profiles were observed. However, this was not the case for papain-like enzyme generated hydrolysates. They also found that the extent of dipeptidyl peptidase-IV (DPP-IV) inhibitory activity, while enzyme-dependent, was not influenced by the hydrolysis conditions. It was also demonstrated that the hydrolysis conditions could yield different kinetics of peptide release with a higher reaction rate occurring during ST in comparison to FF conditions [14]. This may explain the differences in the bioactivity, i.e., antioxidative potencies, of the WPHs generated under ST and FF conditions herein.

The effect of the drying process on the bioactive properties of protein hydrolysates has also been reported elsewhere. SD was reported to enhance the hydroxyl radical scavenging activity and reducing power of silkworm pupae protein hydrolysates, which was suggested to be linked to chemical reactions, e.g., the Maillard reaction, whereas FD did not alter the reducing power [42]. A number of studies have reported an enhancement of the antioxidant properties of whey protein-derived conjugates as a result of the Maillard reaction. For instance, the conjugation of WPC hydrolysates with carrageenan showed a significant (*p* < 0.05) increase in ORAC value (up to 710 µmol TE/g FD powder) compared to non-conjugated samples (~270 µmol TE/g FD powder) [22]. Conjugates of WPI and WPI hydrolysates with galactose also exhibited significantly (*p* < 0.05) higher reducing power, ABTS^•+^ and hydroxyl radical scavenging activities compared to non-conjugated WPI and WPH samples [21]. The authors suggested that a combination of effects between the melanoidins generating through the Maillard reaction and the release of free amino groups during enzymatic hydrolysis may contribute to an enhancement of the antioxidant activity. In addition, WPI-inulin conjugates showed significantly (*p* < 0.05) higher ABTS^•+^ scavenging activity and ORAC values compared to a non-conjugated WPI sample. This was associated with advanced Maillard reaction (browning) products, which can act as hydrogen donators in the free radical scavenging process [20]. Pyo et al. [23] reported that the conjugation of WPC with glucose exhibited a protective effect against oxidatively stressed liver cells (HepG2) through a nuclear factor erythroid 2-related factor 2 (Nrf2)-dependent antioxidant pathway by upregulating antioxidant enzymes. They suggested that this may be linked to an increase in the level of Maillard reaction products, i.e., furosine and hydroxymethylfurfural (HMF) associated with the WPC-glucose conjugate.

SD was not only reported to enhance bioactive properties but also improve powder properties. For example, in the case of soybean hydrolysates, SD reduced the hygroscopicity and increased the glass transition temperature (Tg), which could ultimately lead to better stability at room temperature, in comparison to the FD powder [34]. However, SD-egg white protein hydrolysates resulted in a reduction in its foaming and emulsification properties [33]. On the other hand, a number of previous studies reported similar bioactive properties in protein hydrolysate powders obtained following SD and FD. Wang et al. [34] reported comparable TEAC activity in SD- and FD-soybean hydrolysates. Ma et al. [30] reported comparable proliferative activity in concanavalin A-stimulated splenocytes when treated with SD and FD Alcalase^®^ hydrolysates of WPC80. Similarly, no significant impact of SD vs. FD on FRAP and 2,2-diphenyl-1-picrylhydrazyl (DPPH) scavenging activity of sodium caseinate following stimulated gastrointestinal digestion was reported [31]. In the case of egg white protein hydrolysates, the antioxidant activities tested using DPPH scavenging activity, reducing power and lipid peroxidation were not significantly influenced by the drying process [33]. Therefore, the drying process seems to have an impact on the bio-and technofunctional properties of dried powers in both a substrate- and an enzyme-dependent manner.

## 4. Conclusions

The present study demonstrated the contribution of enzyme preparation (Alcalase^®^ and Prolyve^®^), the hydrolysis and the drying conditions employed on the physicochemical and in vitro antioxidative characteristics of WPHs. The choice of enzyme and the hydrolysis conditions affected the physicochemical and bioactive properties of the resultant hydrolysates, while the drying process had a major impact on the moisture content, powder morphology and the in vitro antioxidative properties. Based on the present results, it can be concluded that SD is a more suitable process to obtain more potent antioxidative ingredients when compared with the use of FD. Nonetheless, the effect of the hydrolysis conditions on the kinetics of WPC hydrolysis (particularly in the case of Prolyve^®^), the bioactive compounds responsible for the antioxidative activities as well as the techno-functional properties of the hydrolysates generated merits further investigation. In addition, the performance of other in vitro and cell-based bioactive assays can contribute to the validation of the outcomes reported herein. Furthermore, identification of the compounds responsible for the higher antioxidative activity in the SD samples requires further investigation.

## Figures and Tables

**Figure 1 antioxidants-11-00399-f001:**
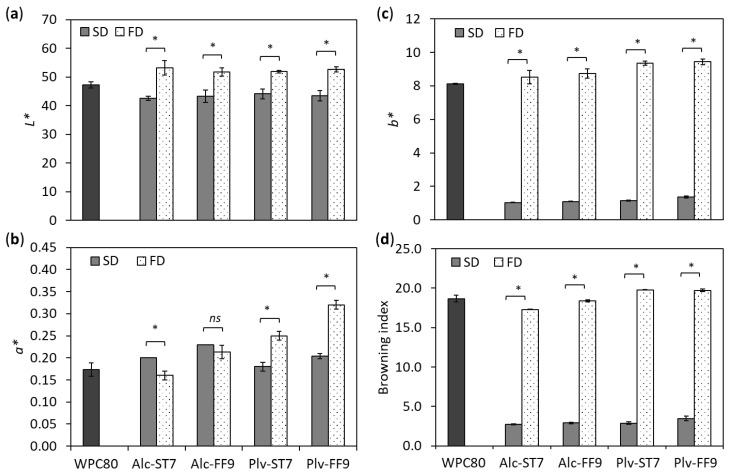
Colour parameters (**a**) lightness (*L**), (**b**) redness (*a**), (**c**) yellowness (*b**) and (**d**) browning index of whey protein concentrate (WPC80) and spray-dried (SD) and freeze-dried (FD) whey protein hydrolysates prepared using Alcalase^®^ (Alc) and Prolyve^®^ (Plv) obtained under pH-stat at pH 7.0 (ST7) and free-fall pH 9.0 (FF9) hydrolysis conditions. Results represent mean ± SD (*n* = 3). * denotes significant difference at *p* < 0.05; *ns* = no significant difference (*p* > 0.05).

**Figure 2 antioxidants-11-00399-f002:**
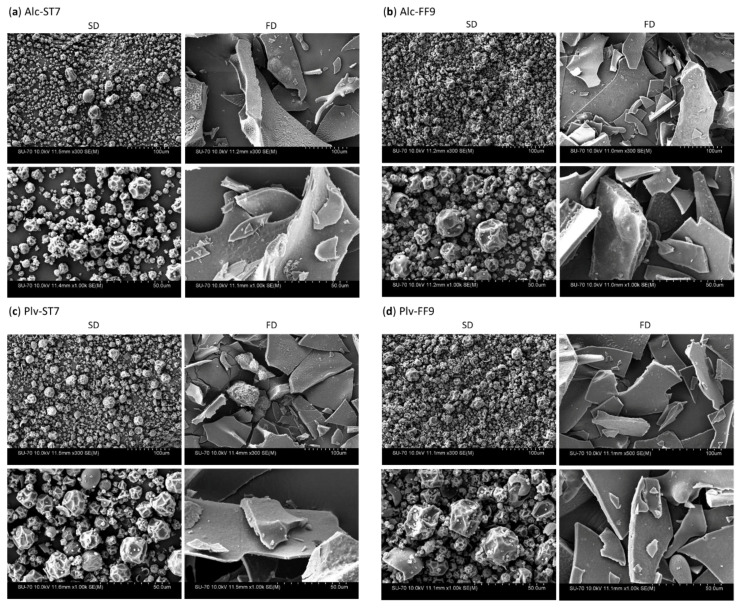
Scanning electron micrographs of whey protein hydrolysed using (**a**) Alcalase^®^ under pH-stat at pH 7.0 (Alc-ST7), (**b**) Alcalase^®^ under free-fall pH 9.0 (Alc-FF9), (**c**) Prolyve^®^ under pH-stat at pH 7.0 (Plv-ST7) and (**d**) Prolyve^®^ under free-fall pH 9.0 (Plv-FF9) where powders were obtained from spray-drying (SD) and freeze-drying (FD).

**Figure 3 antioxidants-11-00399-f003:**
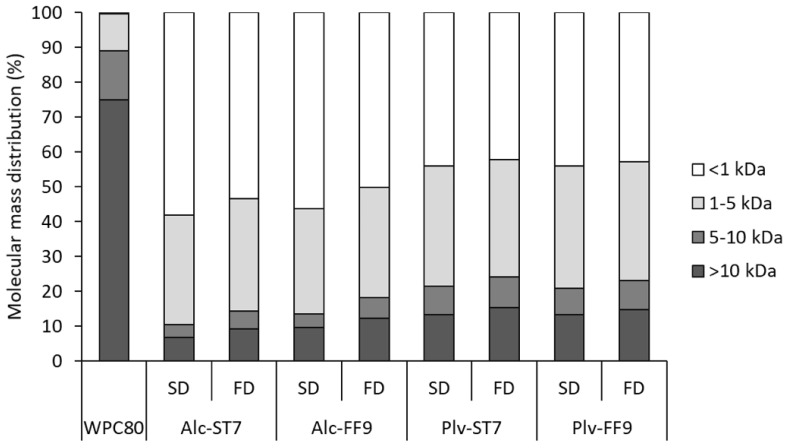
Molecular mass distribution profile of whey protein concentrate (WPC80) and spray-dried (SD) and freeze-dried (FD) whey protein hydrolysates prepared using Alcalase^®^ (Alc) and Prolyve^®^ (Plv) obtained under pH-stat at pH 7.0 (ST7) and free-fall pH 9.0 (FF9) hydrolysis conditions.

**Figure 4 antioxidants-11-00399-f004:**
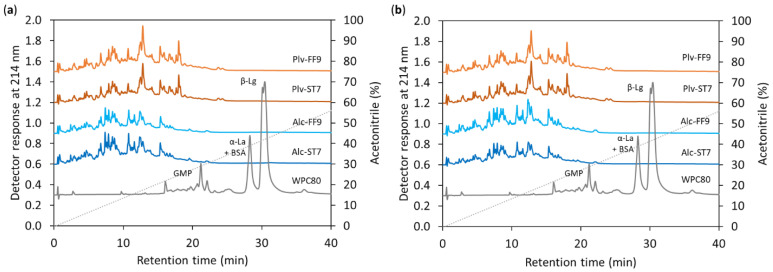
Reverse-phase ultra-performance liquid chromatographic profiles of whey protein concentrate (WPC80) and (**a**) spray-dried and (**b**) freeze-dried whey protein hydrolysates prepared using Alcalase^®^ (Alc) and Prolyve^®^ (Plv) obtained under pH-stat at pH 7.0 (ST7) and free-fall pH 9.0 (FF9) hydrolysis conditions. β-Lg: β-lactoglobulin; α-La: α-lactalbumin; BSA: bovine serum albumin; GMP: glycomacropeptide.

**Figure 5 antioxidants-11-00399-f005:**
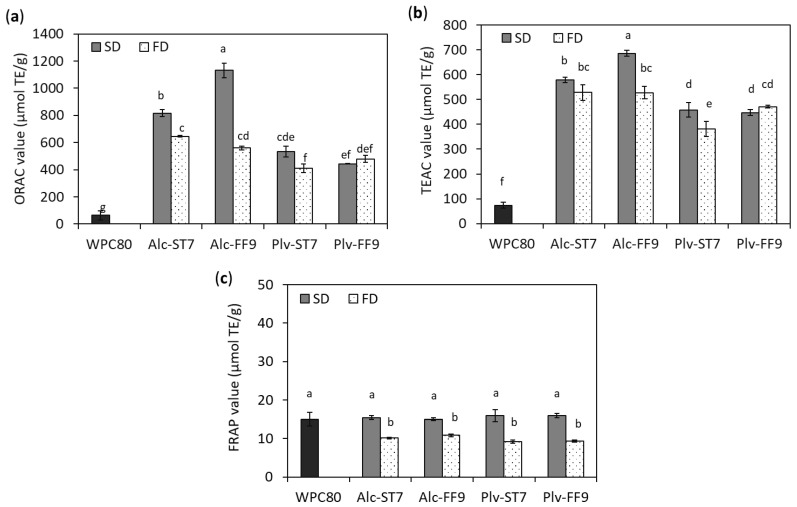
In vitro antioxidative activities, (**a**) oxygen radical absorbance capacity (ORAC), (**b**) Trolox equivalent antioxidant capacity (TEAC) and (**c**) ferric reducing antioxidant power (FRAP) of whey protein concentrate (WPC80) and spray-dried (SD) and freeze-dried (FD) whey protein hydrolysates prepared using Alcalase^®^ (Alc) and Prolyve^®^ (Plv) obtained under pH-stat at pH 7.0 (ST7) and free-fall pH 9.0 (FF9) hydrolysis conditions. Results represent mean ± SD (*n* = 3), where different letters indicate significant difference at *p* < 0.05.

**Table 1 antioxidants-11-00399-t001:** Proximate composition, water activity (a_w_) and the degree of hydrolysis (DH) of spray-dried whey protein hydrolysates (WPHs) generated with Alcalase^®^ (Alc) and Prolyve^®^ (Plv) under pH-stat (ST7) and free-fall (FF9) conditions compared to whey protein concentrate (WPC80).

WPHs	Protein/Protein Equivalent (g/100 g dw)	Ash (g/100 g dw)	Moisture (g/100 g dw)	a_w_ at 20 °C	DH (%)
Alc-ST7	77.09 ± 0.45 ^a^	6.18 ± 0.06 ^c^	5.39 ± 0.12 ^a^	0.335 ± 0.020 ^a^	12.88 ± 0.01 ^b^
Alc-FF9	79.24 ± 1.12 ^b^	5.31 ± 0.35 ^b^	6.25 ± 0.52 ^a^	0.365 ± 0.005 ^ab^	9.52 ± 0.39 ^a^
Plv-ST7	78.21 ± 0.30 ^ab^	5.41 ± 0.09 ^b^	5.57 ± 0.44 ^a^	0.405 ± 0.009 ^cd^	13.24 ± 0.45 ^b^
Plv-FF9	79.34 ± 0.48 ^b^	5.68 ± 0.04 ^bc^	5.77 ± 0.26 ^a^	0.432 ± 0.010 ^d^	12.45 ± 1.27 ^ab^
WPC80	81.82 ± 0.10 ^c^	2.88 ± 0.14 ^a^	4.19 ± 1.13 ^a^	0.388 ± 0.002 ^bc^	0

Values reported as mean ± SD (*n* = 3). Different letters indicate a significant difference between samples (*p* < 0.05; One-way ANOVA). dw: dry weight; a_w_: water activity; DH: degree of hydrolysis.

## Data Availability

Data is contained within the article and Appendix A.

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
