# Peer review of "Contribution of Hydrolysis and Drying Conditions to Whey Protein Hydrolysate Characteristics and In Vitro Antioxidative Properties"

_antioxidants, 2022, doi:10.3390/antiox11020399_

Round 1
Reviewer 1 Report
This study provides important additional information on the most appropriate method of obtaining WPHs. The research methods are modern, the results are correctly analyzed and interpreted, and the conclusions are well underlined.
Introduction
The authors consulted recent bibliographical references to substantiate their research. Although the scientific literature is extremely rich in WPHs and their use in various biotechnology fields, the authors have selected sufficient relevant information.
Materials and Methods
The methods chosen are adequate to evaluate comparatively the two methods of obtaining: spray- vs freeze-drying on the characteristics of whey protein hydrolysates generated under different hydrolysis conditions. References are always quoted.
Results and Discussion
The results are interpreted statistically, are presented correctly graphically and interpreted in correlation with the recent scientific literature. The study, although not addressing a new topic, may be relevant either in research or in industry, in choosing the most appropriate variant of whey protein hydrolyzate depending on the objective.
Author Response
Dear Reviewer,
Thank you very much for your positive comments. We appreciate the time and effort that you have dedicated to providing valuable feedback and insightful comments. The changes made throughout the revised manuscript which are highlighted in yellow.
We provide a point-by-point response to your comments in the attached file.
We look forward to hearing from you in due course.
Sincerely,

Reviewer 2 Report
This manuscript using spray- (SD) and freeze-dried (FD) whey protein hydrolysates from protease hydrolyzed whey protein concentrate to investigate their physicochemical and antioxidative properties. Although the variables of the study are not many, this article is well written.
The following specific comments should be considered while revising the manuscript:
- The analysis method does not write degree of hydrolysis (DH) test method and definition, it needs to be written in Materials and methods section.
- In fig.4, the degree of hydrolysis is about 9-13%, but why the peaks of β-lactoglobulin (β-Lg), α-lactalbumin (α-La), bovine serum albumin (BSA) and glycomacropeptide (GMP) disappeared after hydrolysis.
Author Response

(The authors gave the same response as above.)

Reviewer 3 Report
The paper “Contribution of Hydrolysis and Drying Conditions to Whey Protein Hydrolysate Characteristics and In Vitro Antioxidative Properties” by Kleekayai et al. deals with the effect of hydrolysis conditions and drying process on the properties (morphological, physicochemical and antioxidative) of whey protein hydrolysates.
The paper is clear and well written. The conclusions are strongly supported by data, and the authors performed a thorough analysis of previous literature in the field.
I think the manuscript could be published after the following minor points have been taken into account:
page 3, lines 125-128: how was the freeze drying process performed? What temperature/pressure conditions were used, and what was the duration of the process?
Page 3, line 143: maybe ‘scape’ should be ‘space’?
Table 1 caption: “[…] under pH-stat (ST7 and free-fall (FF9) […]” a parenthesis is missing
Results, lines 188-190: the difference in moisture content may also be due to the freeze drying conditions employed (high temperature/long times). This should be mentioned, I am sure that by adjusting process conditions it would be possible to obtain the same moisture content in SD/FD.
Page 6, line 247: “[…] due the formation of […]” a ‘to’ is missing here
Page 7, line 268: ‘phenomena’ should be ‘phenomenon’
Page 11, line 400: please explain what you mean by GE activity
Page 11, line 441: “[…] compared to The FF […] “ ‘the’ should start with small t
Figure 1 caption: “ns = no significantly difference (p > 0.05).” maybe the authors meant ‘no significantly different’ or ‘no significant difference’?
Author Response
Dear Reviewer,
Thank you very much for your comments. We appreciate the time and effort that you have dedicated to providing valuable feedback and insightful comments. The changes made throughout the revised manuscript which are highlighted in yellow.
We provide a point-by-point response to your comments in the attached file.
We look forward to hearing from you in due course.
Sincerely,
